# Propolis: A Natural Substance with Multifaceted Properties and Activities

**DOI:** 10.3390/ijms26041519

**Published:** 2025-02-11

**Authors:** Simona Martinotti, Gregorio Bonsignore, Elia Ranzato

**Affiliations:** Dipartimento di Scienze e Innovazione Tecnologica (DiSIT), University of Piemonte Orientale, 15121 Alessandria, Italy; simona.martinotti@uniupo.it (S.M.); gregorio.bonsignore@uniupo.it (G.B.)

**Keywords:** anti-inflammatory role, anticancer properties, cardiovascular disease, metabolic syndrome, propolis, rheumatoid arthritis

## Abstract

Propolis (bee glue) is a complex mixture of resins, waxes, and gums, and it is a resinous exudate manufactured by honey bees to maintain the integrity of the hive and defend against external threats. This multifunctional material exhibits several striking properties. The anti-inflammatory properties of propolis have made it a subject of traditional medicine over time, from ancient Egyptian mummification to modern complementary medicine. Propolis with rich phytochemicals, such as polyphenols and flavonoids, exhibit anti-inflammatory, antioxidant, and anticancer effects. This review describes multiple properties and uses of propolis, highlighting the role of propolis as an exceptional natural resource with high therapeutic potential.

## 1. Introduction

Honey bees create propolis, a naturally occurring sticky substance commonly referred to as bee glue, by combining saps, resins, and mucilages extracted from different plant components, including flower buds, tree barks, and leaves, with beeswax and a number of bee enzymatic proteins [1].

Honey bees utilize this natural substance to repair damage to the hive, polish the interior walls, and keep the hive’s humidity and temperature steady. Additionally, the colony is protected against parasites, predators, and disease microorganisms by its use [2]. Propolis becomes stiff and brittle at low temperatures and soft, flexible, and extremely sticky at high temperatures [3]. Depending on the source and storage period, propolis can have distinct herbaceous fragrances and come in a range of hues, including brown, yellow, green, and red [1].

Across generations and cultures, propolis’s medicinal properties have been thoroughly investigated in traditional medicine [1]. Because it inhibits the growth and decomposition of bacteria and fungi, it was primarily employed by the ancient Egyptians to embalm their cadavers. Hippocrates used propolis to cure wounds and ulcers. Additionally, in the 17th century, propolis was included as a legitimate treatment in the British Pharmacopoeia. During World War II, propolis was used as an antibacterial and anti-inflammatory medication [4]. As a result, propolis is now found in many supplementary healthcare products, including toothpaste, throat sprays, cough syrups, tinctures, mouthwash preparations, gels, skin lotions, shampoos, creams, lozenges, and chewing gum [5].

Propolis has biological qualities such as antioxidant, anti-inflammatory, and anti-tumor potential due to the presence of phytochemicals (polyphenols, phenolic aldehydes, flavonoids, ketones, and phenolic acids) [1,6,7] (see Figure 1).

## 2. Chemistry of Propolis

Propolis has a chemical makeup that varies depending on its botanical and geographic origin, including climate, plant resources, location, and the time of bee collection [8]. Because the resins that are a vital component of propolis are soft and malleable, during the warmest parts of bright days, honey bees collect plant material for the synthesis of propolis.

Thus, propolis production occurs in temperate regions from late summer to fall, while honey bees can gather plant material all year round in tropical regions [9]. The primary determinant of propolis’s chemical makeup and, consequently, its biological and pharmacological characteristics is the distinctiveness of the local flora.

The kind of solvent employed during the extraction process affects the propolis extracts’ chemical makeup and biological activity. Ethanol is the most often utilized solvent for propolis extraction, especially when it is 70–75% concentrated [10]. Propolis can be extracted using a variety of solvents, including water, ethyl ether, methanol, hexane, chloroform, glycolic and glyceric solution, and seed oil [11]. Propolis is actually incorporated into pharmaceutical and medical products as aqueous and ethanolic extracts [11].

Propolis typically consists of 50–60% resins and balms, 30–40% waxes and fatty acids, 5–10% essential and aromatic oils, 5–10% pollen, and roughly 5% additional materials, such as vitamins A, B complexes, C, and E, as well as vital minerals, including calcium, salt, potassium, aluminum, copper, magnesium, iron, and zinc [12] (see Figure 2). Based on data from the literature, propolis samples from various geographic locations have been found to contain over 300 distinct chemicals [10,12].

Alkaloids, including chrysin, pinocembrin, apigenin, galangin, kaempferol, quercetin, cinnamic acid, o-coumaric acid, p-coumaric ac-id, caffeic acid (CA), and caffeic acid phenylethyl ester (CAPE), along with flavonoids, aliphatic and aromatic acids, phenolic esters, fatty acids, alcohols, terpenes, β-steroids, and alkaloids, are all found in propolis [13].

The primary compounds that give propolis its pharmacological characteristics are flavonoids, while terpenoids also give propolis its smell [3]. Propolis’s biological activities are the outcome of interactions between different chemicals (see also Table 1).

## 3. Antioxidant Properties

Propolis’s antioxidant qualities have been thoroughly examined and demonstrated using several techniques [14]. Propolis extracts were found to have an antioxidant capacity comparable to that of ascorbic acid or butylated hydroxytoluene, two synthetic antioxidants, in the same in vitro tests [15].

Propolis’s antioxidant capacity depends on its content; yet, research attempting to identify clear correlations between these two factors has not been consistent [16].

Propolis extracts’ total phenolic content, as reported in the literature, generally ranged between 30 and 200 mg of gallic acid equivalents (GAEs)/g of dry weight, while their flavonoid content ranged between 30 and 70 mg of quercetin equivalents (QEs)/g. Meanwhile, their free radical-scavenging activity ranged between 20 and 190 μg/mL [16,17,18]. It is thought that the phenolic compounds—which are distinct from flavonoids—are what give Brazilian propolis its antioxidant properties.

In contrast to Brazilian propolis, poplar propolis’s antioxidant activity seems to be largely influenced by both its total polyphenol and total flavonoid contents [17]. Fabris et al.’s results [19] showed that propolis samples from Europe (Italy and Russia) had similar polyphenolic compositions and, therefore, similar antioxidant activity, while Brazilian propolis had lower amounts of polyphenols and, therefore, fewer antioxidant properties. Overall, there appears to be a significant issue with standardizing propolis composition because it is highly dependent on a variety of factors, including bee species, plant origin, geographic location, temperature variation, seasonality, and storage conditions [17].

The majority of research on propolis’s antioxidant qualities has been conducted on animals or cell cultures. Few studies examining propolis’s antioxidant effect in humans can be found in the literature that is currently available.

The effects of oral administration of commercially available propolis solution twice daily for 90 days (15 drops each) on the oxidative state and lipid profile in a Chilean population were assessed by Mujica et al. [20]. The scientists came to the conclusion that using propolis supplements seemed to increase HDL and oxidative state, which may lower the risk of cardiovascular events.

In turn, Jasprica et al. [21] looked into the potential impact of 30-day supplementation with commercially available powdered propolis extract (a total daily dose of flavonoids was 48.75 mg) on antioxidant enzymes in healthy individuals. The authors came to the conclusion that propolis had a time- and gender-dependent effect on lipid peroxidation, and they also suggested that there might be a temporary effect of propolis ingestion on lipid peroxidation.

Zhao et al. [22] investigated the impact of Brazilian green propolis supplementation on antioxidant status in patients with type 2 diabetes mellitus. The propolis administration (900 mg/day, 18 weeks) was linked to a decrease in serum carbonyls (protein oxidation markers) and a rise in GSH. Additionally, the Brazilian green propolis group showed a significant increase in IL-1β and IL-6 sera levels and a decrease in TNF-α serum levels; however, the propolis treatment had no effect on serum glucose, glycosylated hemoglobin, insulin, aldose reductase, or adiponectin. These findings suggest that propolis affects oxidative stress in type 2 diabetic patients but not diabetes parameters.

The in vitro and animal studies appear to support the value of using propolis as a natural agent that can counteract the negative effects of oxidative stress, which underlies the pathogenesis of many diseases or disorders. There are, however, relatively few research studies on their function in humans, and those that have been conducted have primarily focused on assessing the impact of supplementing with commercially available propolis extracts in individuals with type 2 diabetes or in healthy individuals. Therefore, without a thorough chemical analysis, it is challenging to make a broad judgment about their possible therapeutic use.

## 4. Anti-Inflammatory Role

The immune system coordinates the process of inflammation, which is typically brought on by tissue injury and/or microbial infections. If the triggers result in the reduction and resolution of inflammatory processes, which are controlled by anti-inflammatory cytokines and other mediators, the inflammatory response is deemed successful.

Chronic inflammatory milieu marked by ongoing leukocyte infiltration and increased amounts of pro-inflammatory cytokines shift the homeostatic baseline [23].

Injurious inflammation, on the other hand, happens if the harmful inflammatory triggers cannot be eliminated. Propolis may help control the elements of the immune system because of its ability to modulate inflammation. The majority of research on propolis’s impact on immune system components has been conducted both in vitro and ex vivo. Propolis seems to modulate innate immunity by reducing inflammation. Propolis decreased the expression of inflammatory cytokines like IL-1α, IL-1β, IL-4, IL-6, IL12p40, IL12p70, IL1-3, monocyte chemoattractant protein-1 (MCP1), and granulocyte-macrophage colony-stimulating factor (GM-CSF) in LPS-activated peritoneal macrophages isolated from C57BL6 mice [24]. It is interesting to note that propolis decreased the expression of some inflammatory genes while raising the expression of others, indicating that propolis has complex immune-modulatory qualities that require more research [24].

Propolis has been shown in numerous in vivo investigations to have an impact on immune system modulation toward regulatory profiles and anti-inflammatory environments. For example, in the peritoneal cavity and visceral adipose tissue of lean and obese mice, propolis promoted the trans-differentiation of M1 macrophages to D11b+, Gr-1+ myeloid-derived suppressor cells (MDSCs). The result of this trans-differentiation was anti-inflammatory [25].

Propolis was shown [26] to reduce the rise in neutrophil infiltration brought on by intraperitoneal injection of monosodium urate (MSU). Additionally, in peritoneal lavage fluid, propolis suppressed the production of MSU-induced IL-1β, active caspase-1, IL-6, and MCP-1 [26].

Additionally, propolis and its bioactive components, particularly artepillin C, enhanced TNFR2 expression in Tregs in mice via the IRF4/cMyc axis, indicating that propolis may be able to up-modulate Tregs to reduce inflammatory disorders brought on by auto-immune problems [27].

Propolis inhibits the IgE/antigen-induced production of IL-4, IL-6, and IL-13 in basophils in a mouse model of allergic inflammation. In basophils treated with propolis, phosphorylation of the FcεRI signaling molecules Lyn, Akt, and ERK was suppressed. Additionally, propolis reduces intestinal anaphylaxis caused by basophils and basophil-derived IL-4 and down-regulates IgE-mediated chronic allergic inflammation (IgE-CAI) [28].

Propolis has been shown in numerous studies to have various pro-inflammatory properties in immunosuppressive environments, indicating that it may have immunomodulatory and/or immune-restoring properties. According to Oliveira [11], propolis (obtained with maceration in 70% ethanol at room temperature under moderate shaking) may be able to counteract the immunosuppressive effects of the chemotherapeutic medication doxorubicin. In monocytes exposed to doxorubicin, propolis boosted the production of TLR-4, TNF-α, NF-κB, and IL-10. It is noteworthy that propolis decreased doxorubicin-exposed monocytes’ phosphorylation of IκBα and IL-1β expression [29]. Additionally, in monocytes that had been immunosuppressed by docetaxel, propolis restored the expression of IL-6, TNF-α, and HLA-DR [30].

Acute and/or chronic inflammation in the kidney, lung liver, brain, cardiovascular system, gastrointestinal tract, and reproductive organs can be caused by inflammatory reactions that are triggered by a number of factors, including toxic compounds, pathogens, and damaged cells. This can result in long-lasting cellular and tissue damage [31]. Propolis has the potential to be used therapeutically in this area because of its anti-inflammatory qualities. Medjeber and co-workers [32] discovered that propolis (ethanolic extract) markedly increased the expression of IL-10 while considerably inhibiting the expression of nitric oxide and interferon-γ in peripheral blood mononuclear cells isolated from patients with celiac disease. Additionally, propolis decreased the activity of NF-κB and pSTAT-3 transcription factors, as well as the production of iNOS [32].

### 4.1. Metabolic Syndrome

Metabolic syndrome (MeS) is a prevalent, complex illness. Abdominal obesity, hypertension, impaired fasting blood glucose, and dyslipidemia are among the pathological characteristics that describe it. Individuals with MeS are susceptible to a number of concomitant conditions, including type 2 diabetes, cancer, cardiovascular disease, and polycystic ovarian syndrome [33]. Additionally, MeS is strongly linked to the emergence of inflammatory conditions and oxidative stress. MeS is recognized as a global health hazard. According to reports, MeS places a significant financial strain on the healthcare system, and its risk factors can lower people’s quality of life and mental and physical health.

Propolis may be able to help reduce chronic inflammation, which is one of the characteristics of metabolic syndrome.

Propolis has been shown in numerous research on both humans and animals to be beneficial in lowering blood pressure, controlling lipid and glucose metabolisms, and boosting immune system performance [34]. Nevertheless, some studies yield contradictory findings. It could be because of variations in the form, amount of propolis utilized, and length of study. Propolis dramatically increased fasting plasma glucose but had no effect on serum insulin or homeostatic model assessment insulin resistance (HOMO-IR) [35], according to the results of a meta-analysis that comprised six studies. A clinical investigation found that giving diabetic patients 1000 mg/d of Iranian propolis for 90 days dramatically decreased inflammatory biomarkers and HOMO-IR while simultaneously raising HDL-C levels as compared to controls [36].

The vast majority of the metabolic syndrome-related research that was found and included in this evaluation were in vivo animal models. Sakai and co-workers [37] showed that in mice given a high-fat diet, propolis consumption dramatically decreased the infiltration of T lymphocytes and macrophages into the epididymal fat [37]. In mice fed a high-fat diet, propolis was also demonstrated to decrease insulin resistance, body weight increase, and fat formation. Furthermore, propolis increased the expression of the anti-inflammatory IL-10 in hepatic and adipose tissues while drastically lowering the dose-dependent mRNA expression level of pro-inflammatory cytokines TNF-α, IL-1β, and IL-6 [38].

Propolis’s capacity to lower the amount and expression of pro-inflammatory cytokines in conditions linked to metabolic syndrome has been validated by numerous animal investigations. But when Cardinault and colleagues [39] compared three different types of propolis (prepared as ethanolic extracts)—poplar, Dalbergia, and Baccharis—they discovered that while Baccharis and Dalbergia propolis seemed to increase the expression of mRNAs coding for TNF-α, chemokine C-C motif ligand 5 (Ccl5), and Ccl2, poplar propolis decreased these mRNAs in mice fed a high-fat diet. Remarkably, the polyphenol profiles of Baccharis propolis and poplar showed similarities, indicating that other kinds of bioactive substances might be responsible for this variation in impact [39].

### 4.2. Type 2 Diabetes Mellitus

The medical illness known as type 2 diabetes mellitus (T2DM) is brought on by an increased blood sugar (glucose) level, which is brought on by the body’s inability to produce enough insulin [40]. Research on natural chemicals as a preventative measure for type 2 diabetes has garnered a lot of interest. Propolis, which includes flavonoids that serve as antioxidants, anti-inflammatory agents, and free radical scavengers, has been shown in a few trials to offer potential therapeutic benefits for the treatment of complicated disorders, including type 2 diabetes [41]. In fact, propolis contains apigenin, chrysin, galangin, kaempferol, luteolin, genistein, and pinocembrin, all of which have been shown to have antidiabetic properties [42].

In addition to its anti-inflammatory, antioxidant, anticancer, antiapoptotic, and antiosteoporosis qualities, naringin, a naturally occurring flavanone glycoside present in propolis, has been shown to have lipid-reducing and insulin-like qualities that lower insulin resistance and hyperglycemia [42].

A few recent studies have also shown that propolis use has a substantial impact on T2DM patients, lowering their blood glucose, serum insulin, and serum glycosylated hemoglobin (HbA1c) levels [36]. According to some theories, propolis may affect glucose metabolism by inhibiting intestinal α-glucosidase activity during the digestion of carbohydrates and by activating the β-cells in the pancreatic islets of Langerhans, which would raise insulin secretion [43]. According to Zakerkish et al., propolis administration for 12 weeks reduced insulin resistance and insulin levels in T2DM patients [36]. Numerous studies have also demonstrated that oxidative stress, inflammatory cytokines, and free radicals have a significant impact on the onset and consequences of T2DM [44].

Important organs like the heart, kidneys, nerves, and eyes thereafter sustain oxidative damage as a result of T2DM linked to ROS [42]. Metabolic syndrome traits and hyperglycemia are linked to subclinical inflammation, which results in insulin resistance. Elevated glucose levels cause peripheral tissue to absorb glucose and reduce intestinal carbohydrate absorption [43]. Propolis’s antihyperglycemic properties stop the synthesis of glucose from dietary carbohydrates, and it can improve insulin resistance and regulate a spike in postprandial glucose [43]. So, propolis can be a promising treatment for diabetic mellitus prevention and control, improving the glycemic and lipid profiles of T2DM patients.

### 4.3. Rheumatoid Arthritis

The human body’s immune system triggers rheumatoid arthritis (RA), a chronic immuno-inflammatory disease that causes excruciating pain and limits joint function [45]. Oxidative stress and inflammation have a major impact on the pathophysiology and associated consequences of RA. The development of RA is favorably associated with inflammation, which has been linked to ROS-activating nuclear factor kappa B (NF-κB) [46]. The immune response can be influenced by a number of circumstances, which might lead to the release of pro-inflammatory cytokines. These cytokines can trigger synovial cell activation and inflammation, which can alter joint regeneration processes.

Because of their anti-inflammatory and antioxidant properties, natural products like propolis, which may have fewer side effects and are more affordable, have been shown to be successful in treating RA [47]. It has been demonstrated that propolis’s chemical components—specifically, its terpenoids, phenolics, steroids, alcohols, terpenes, and sugars—are what provide the therapeutic effects. Additionally, propolis can reduce ROS by increasing antioxidants and inhibit inflammatory cascades by blocking the NF-κB pathway [45].

Propolis’s chemical constituents possess potent anti-inflammatory qualities that enable them to control the fundamental immune cell function and reduce cytokines triggered by the immune response in T-cells and NF-κB activation [48]. Ansorge et al. pointed out that propolis’s caffeic acid, caffeic acid phenylethyl ester (CAPE), hesperidin, and quercetin can inhibit T cell DNA synthesis and inflammatory development while promoting the cells’ production of the transforming growth factor-β1 (TGF-β1) [49]. According to Zhang et al., galangin and apigenin lower TNF-α mRNA levels [50]. Furthermore, CAPE is an important propolis molecule with anti-inflammatory qualities that also functions as a selective inhibitor of NF-κB activation and can accurately block NF-κB activation by a variety of inflammatory stimuli, such as TNF-α [1,7].

It is currently unknown whether propolis has any negative consequences on RA disease activity. Propolis’s impact on RA patients has been assessed through a multicenter, double-blind, randomized, monitored trial [51]. According to the clinical trials, propolis from stingless bees did not lower disease activity or enhance RA patients’ quality of life. Propolis’s inability to inhibit disease activity was linked to the drugs the individuals were taking before the clinical research. Instead, Brazilian propolis has been shown to help lower the activity of RA disease in mice, suggesting that propolis could offer new therapeutic alternatives [51].

### 4.4. Cardiovascular Disease

One of the main risk factors for the heart and blood vessels globally is cardiovascular disease (CVD), which can result in angina, heart attacks, and strokes [52]. Usually, blood artery constriction or blockage is the reason. It is interesting to note that propolis may help treat cardiovascular conditions like atherosclerosis, hypertension, and ischemia-reperfusion (IR) injury [53]. Risk factors, including obesity and oxidative stress, have a very high probability of affecting CVDs; nevertheless, propolis supplementation, with its bioactive components, may lower the risks of CVD [54].

According to the research, propolis’s cardiovascular effects are often linked to its anti-inflammatory, antihypertensive, antiangiogenic, antiatherosclerosis, antioxidant, and immunomodulatory qualities [55].

The anti-inflammatory properties of propolis have been linked to terpenoids, steroids, amino acids, and polyphenols, including flavonoids, phenolic acids, and their esters [56]. Inhibition of cyclooxygenase, which inhibits the synthesis of prostaglandins, scavenging of free radicals, inhibition of nitric oxide synthesis, reduction of pro-inflammatory cytokine levels, and immunosuppressive activity are the main mechanisms behind propolis’s anti-inflammatory action [47]. The caffeic acid phenethyl ester (CAPE) is one of the primary anti-inflammatory and anticancer compounds present in propolis. As a powerful modulator of arachidonic acid (AA), CAPE stops AA from escaping the cell membrane and stops the synthesis of LOX and COX enzymes, which are essential for the processes that lead to AA metabolism [57]. Additionally, CAPE reduces the production of pro-inflammatory cytokines while increasing the production of two anti-inflammatory cytokines, IL-10 and IL-4 [44].

A major characteristic of many cardiovascular diseases is endothelial dysfunction, which is defined by a reduction in the endothelium’s capacity to emit NO [58]. Propolis’s antioxidant effect on endothelial cells has been shown in numerous investigations, the majority of which used Chinese propolis variants. Chinese poplar propolis provided protection against oxLDL-induced endothelial dysfunction by inhibiting the PI3K/Akt/mTOR signaling pathway and reducing ROS generation and LOX-1/p38 mitogen-activated protein kinase (MAPK) expression [59]. Chinese propolis protects the endothelium against endothelial damage caused by lipopolysaccharides by blocking the MAPK/NFκ-B signaling cascade and inhibiting autophagy [60].

Hemostasis is a set of processes that permits the repair of damaged blood vessels while preventing vascular blockage (i.e., fibrinolytic activity) and maintaining appropriate blood rheology [61]. With several propolis preparations, antiplatelet activity has been seen both in vivo and in vitro. The ethanolic extract of Brazilian green propolis inhibited the production of plasminogen activator inhibitor-1 (PAI-1) in cultured human umbilical cord endothelial cells (HUVECs), which was mediated by TNF-α [62]. Results were shown to be similar in animal studies. An ethanolic extract of Brazilian green propolis was administered to mice for eight weeks, and this prevented the rise in PAI-1 and its plasma activity caused by lipopolysaccharide (LPS). The antiplatelet action of Indonesian propolis was found to lengthen the duration of bleeding in mice by a factor comparable to that of aspirin in an in vivo study [63].

Propolis also prevents protein nitration and LDL peroxidation in vitro, reducing the risk of CVD [41]. Red propolis from Cuba has shown preventative effects in animal models of alcohol-induced liver injury; this is probably due to its antioxidant properties [64].

Immune cells are drawn to and proliferate in the artery wall due to a complicated mechanism involving the accumulation and alteration of plasma lipoproteins. Food-based polyphenols lower the risk of cardiovascular diseases and prevent the development of atheromatous plaques [41]. Therefore, a viable alternative strategy for avoiding cardiovascular illnesses is propolis, a high source of polyphenols. Propolis has been shown to affect the metabolism of lipids and lipoproteins. Propolis has been shown to reduce ox-LDL, which, in turn, reduces ROS and inhibits the activation of NF-κB in HUVEC cells [65]. Ox-LDL is a strong ROS inducer. By triggering downstream signal molecules like NF-B, oxidative stress—which is brought on by an increase in ROS production—can hasten the onset of several diseases, including atherosclerosis [62].

## 5. Anticancer Properties

Cancer is characterized as a condition where cells develop abnormally and have the ability to spread and invade other sections of the body. Certain cancer cells may acquire drug resistance, which can decrease the effectiveness of drugs, making cancer cells resistant to treatments. Although there is no definitive cure for cancer, there are a number of treatments that can slow the growth of tumors. As a result, scientific research has concentrated on creating possible medications from natural resources to treat cancer because the current approaches are, in some cases, ineffective [66,67].

Natural compounds are currently among the most widely used as complementary and alternative medications [68]. This is because human health benefits from a vast array of naturally occurring bioactive chemicals. And the majority of anticancer drugs are made from natural ingredients [66].

Recent years have seen a large body of research on the anticancer effects of propolis, which is produced by numerous bee species found in different geographic locations, in the treatment of cancer cell lines from different tissues, such as breast, colon, liver, lung, and pancreas [69,70].

Propolis’s anticancer properties are greatly influenced by its variety of chemical components. Flavonoids, one of propolis’s active constituents, have chemopreventive properties against the majority of carcinogenesis. Apigenin, caffeic acid, CAPE, ferulic acid, galangin, luteolin, myricetin, pinocembrin, and quercetin are other active propolis components that have anticancer and antiproliferative qualities [69]. Targeting molecules crucial to apoptosis through the intrinsic pathway, propolis also functions as a pro-apoptotic protein, triggering the caspase cascade mechanisms and releasing cytochrome C from the mitochondria into the cytosol [40].

Cell differentiation, proliferation, and migration are regulated by the primordial and evolutionarily conserved Wnt signaling system [71]. APC and Axin are suppressors of the Wnt pathway, and β-catenin, another downstream protein, has a role in the development of cancer. Many malignancies have mutations in the β-catenin gene. In human colon cancer cell line HCT16, CAPE treatment inhibited the Wnt signaling pathway [72]. Chrysin stimulates the p38 mitogen-activated protein kinase activity, which, in turn, inhibits the cell cycle and decreases cell proliferation in C6 gliomas [73]. This results in an increase in p21 protein expression. By raising the levels of endothelial nitric oxide synthases (e-NOS) and inducible nitric oxide synthases (i-NOS), CAPE has cytotoxic and antiproliferative effects on breast cancer cell lines [74]. In 3T3-L1 fibroblast cells, the CAPE treatment resulted in decreased production of cyclin D and inhibited Akt and extracellular signal-regulated kinase (ERK) phosphorylation [75].

In U937 cells (histiocytic lymphoma cells), chrysin administration demonstrated apoptosis induction through PI3K/Akt signaling suppression and NF-κB/inhibitor of apoptosis (IAP) inactivation, which triggered caspase-3 and death [76].

Moreover, AGP-01 gastric cancer cells are susceptible to the cytotoxic effects of Brazilian green propolis. Two of the main substances that support these actions are p-coumaric acid and artepillin C [77]. In cancer-resistant prostate cancer CWR22Rv1 cells, artepillin C, a prenylated derivative of p-coumaric acid, induces apoptosis as shown by DNA fragmentation and increases poly ADP-ribose polymerase and cleaved caspase-3 [34]. A reduction in survivin levels was at least partially responsible for the cell death that artepillin C caused in oral squamous cell carcinoma cells. A member of the inhibitor of apoptosis family, survivin is overexpressed in the majority of human malignancies but not present in healthy adult tissues [78].

Propolis and several of its constituents exhibit antiangiogenic activity against neoplastic cells, according to experimental evidence [79]. By lowering the activation of HIF-1α (hypoxia-inducible factor 1), caffeine can reduce the phosphorylation of JNK-1 (c-Jun N-terminal kinases) and, hence, angiogenesis. These processes have the effect of decreasing vascularization brought on by VEGF (vascular endothelial growth factor), which, in turn, inhibits the growth of tumors. By suppressing metalloproteases MMP2 and MMP-9 and preventing STAT3 (transcription factor and signal translation 3) signaling, caffeic acid also lowers the angiogenesis of hepatocarcinoma cells [80].

The most intricate process that results in the growth of secondary tumors in tissues or organs that are far from the original tumor is metastasis [81]. One important factor in metastasis is the epithelial–mesenchymal transition (EMT) [82]. The activity of signaling pathways essential for metastasis is influenced by propolis and its constituents. Propolis has been shown to slow the growth of tumors in a mouse model of breast cancer. When animals with DMBA (7,12-dimethylbenanthracene)-induced mammary gland cancer are given propolis, the expression of the Wnt2 and FAK (focal adhesion kinase) proteins is suppressed [83].

Through the MAPK and PI3K/AKT signaling pathways, propolis and its constituents can also alter the capacity of cancer cells to move and invade [84,85]. Propolis from East Pacific locations, including Taiwan and Okinawa, contains particular active chemicals called c-prenylflavonones, which include propolin C [84]. In the HCC827 lung cancer cell line, propolin C inhibits ERK and AKT phosphorylation in a dose-dependent manner. Vimentin, a mesenchymal-like cell marker, and Snail, a key inducer of EMT, showed dose-dependent downregulation following propolin C administration, whereas E-cadherin, an epithelial-like cell marker, showed upregulation [84].

Human laryngeal cancer cells (Tu212 and HEP-2 cell lines) treated with galangin have shown a reduced capacity for invasion and migration. Additionally, galangin inhibited p38, AKT, and NF-κB phosphorylation and downregulated the expression of the Ras, Raf, and PI3K proteins [73]. By promoting the expression of NDRG1 (N-myc downstream regulated 1), CAPE inhibits the invasion and EMT of oral squamous cancer (SAS and OECM-1 cell lines) and nasopharyngeal carcinoma (TW01 and TW04 cell lines). By controlling the amounts of N-cadherin, E-cadherin, vimentin, Snail, and Slug proteins, NDRG1 contributes to the inhibition of migration and EMT. It is interesting to note that CAPE treatment of oral and nasopharyngeal cancer cells elevated phosphorylation of p38, JNK, and ERK in a way that was dependent on time and dose [85].

A major part of immunosuppression and carcinogenesis is played by tumor-associated macrophages (TAMs). Co-culturing both types of cells or treating HepG2 (hepatocellular carcinoma cells) with the supernatant of human monocytic THP-1 cells and HT-29 (colorectal adenocarcinoma cells) with the supernatant of M2-like macrophages induced the migration and invasion of the cancer cells. It is noteworthy that treatments with Cuban brown propolis extract reduced invasion and migration, both with and without M2-like macrophages [86]. Pro-inflammatory factors like IL-8, IL-10, CCL2, and VEGF were shown to have lower mRNA levels in M2-like macrophages treated with Cuban propolis.

The biological effects of natural products in cancer treatment have been assessed in a number of studies. Vincristine, vinblastine, and taxanes (paclitaxel and docetaxel) are examples of natural chemicals and their derivatives that are employed as chemotherapeutic medicines. Furthermore, natural substances may reduce the more severe effects of anticancer therapy and shield healthy cells from the harm produced by radiation and chemotherapy [87].

The effects of tamoxifen, CAPE, and their combination on the growth of tumors, survival duration, and lifespan of Ehrlich tumor-bearing mice were investigated by Motawi and associates [88]. Compared to mice treated with either tamoxifen or CAPE alone, tumor-bearing mice treated with a combination of the two drugs had a twofold longer lifespan and considerably reduced tumor size and weight [88].

The efficiency of chemotherapy with cytotoxic drugs may also be impacted by propolis. When compared to either propolis or 5-FU alone, Sameni and colleagues showed that administering Iranian propolis extract, along with 5-fluorouracil (5-FU), significantly reduced the incidence of aberrant crypt foci brought on by azaxymethane in a mouse model of colorectal cancer. Additionally, the expression of Cox-2, iNOS, and β-catenin proteins—all of which are crucial for the occurrence and development of colorectal cancer—was reduced by the propolis and 5-FU combination [89].

Another major issue in cancer treatment is multidrug resistance (MDR). MDR is the biological process via which cancer cells in patients become resistant to unrelated chemotherapeutic drugs [90]. Chemotherapeutics are effluxed from cancer cells by the multidrug membrane transporter P-glycoprotein (P-gp). According to Kebsa et al., propolis increased the intracellular content of doxorubicin and dose-dependently blocked the P-gp efflux pump [91]. By suppressing P-gp expression, quercetin, ferulic acid, and CAPE may also affect the MDR of neoplastic cells [90].

One of the most common side effects of radiation and chemotherapy is oral mucositis [92]. Propolis was safe and well-tolerated by breast cancer patients undergoing chemotherapy [93]. During chemotherapy, individuals with breast cancer experienced a significant decrease in oral mucositis symptoms when they rinsed their mouths with a dry extract of propolis. Patients with head and neck cancer undergoing treatment also saw comparable outcomes [94].

In general, in clinical trials, propolis is typically well tolerated by cancer patients [93].

To develop new cancer treatments and address the issue of chemotherapy resistance, more research is needed on the bioactive chemicals of propolis and the target molecules involved.

## 6. Conclusions

Propolis, being a naturally occurring material, lacks a consistent and repeatable chemical makeup. As a result, identifying its complete makeup in a trustworthy manner presents a significant research challenge. As a result, standardization techniques are required in order to combine the presence of particular chemicals with biological activity and to create guidelines for the application of various propolis varieties [95].

Relatively little research has examined propolis’s bioavailability or the quantity of propolis and/or its elements that can be absorbed into a living organism and reach the sites where they may exert their biological effects through systemic circulation [96]. It is challenging to assess this parameter since it can be influenced by a wide range of elements, including the food matrix, potential interactions with other substances, the environment, and chemical structure and concentration of propolis components.

Therefore, more clinical studies are required to confirm the proportion of propolis chemicals in human individuals. Additionally, propolis allergy reactions and dosage merits need to be investigated [93].

## Figures and Tables

**Figure 1 ijms-26-01519-f001:**
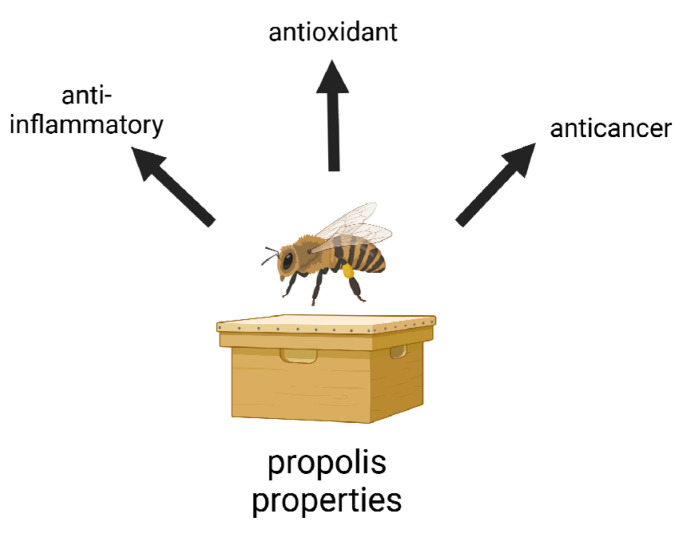
Main known properties of propolis. For further information, see the text. Created in BioRender (https://biorender.com/x11y323, accessed on 9 February 2025).

**Figure 2 ijms-26-01519-f002:**
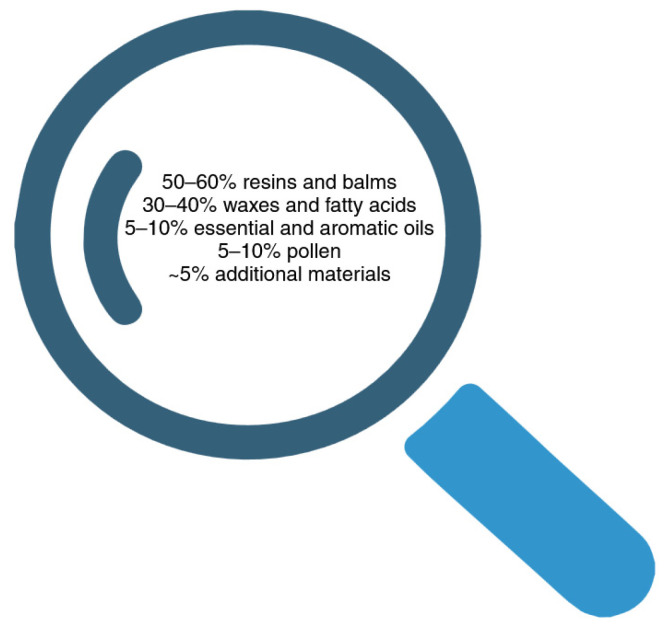
Typical propolis composition. For further information, see the text. Created in BioRender (https://biorender.com/p26p562, accessed on 9 February 2025).

**Table 1 ijms-26-01519-t001:** Propolis main components.

**Resin**	In the hive, the bees collect plant resins and use them as polishers, sealants, disinfectants, and mummifiers for dead insects.
**Wax**	Typically, honey bees generate the yellowish, soft, and highly absorbable substance known as wax. Esters, acids, high-fat alcohols, and even free hydrocarbons are all found in waxes. Although wax is a durable and extremely moisture-proof material, it is not resistant to mechanical stresses or heat.
**Flower pollen**	The flower from which the honey bee gathers its pollen determines its precise composition. Essential amino acids, vitamins, mineral salts, and hormones are abundant in flower pollen.
**Phenols**	Flavonoids, phenolic acids, tannins, stilbenes, curcuminoids, coumarins, and quinines are all found in propolis. The proportion of these types of substances varies and depends on the place and time of collection. Strong antioxidant and antibacterial properties were due to the high total phenolic and flavone/flavonol contents in the propolis.
**Terpenes**	Primary and secondary metabolites are produced by all plants and serve a variety of purposes. Amino acids, simple sugars, nucleic acids, and lipids are all found in primary metabolites and are necessary for cellular functions. Terpenes, alkaloids, and phenolic compounds are among the substances found in secondary metabolites that are created in reaction to stress. Terpenes are also responsible for propolis’s distinctive resinous smell.
**Hydrocarbons**	Propolis from various geographical regions has been found to contain alkanes, alkenes, alkadins, monosasters, diesters, aromatic esters, fatty acids, and steroids.

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
