# Peer review of "Propolis: A Natural Substance with Multifaceted Properties and Activities"

_ijms, 2025, doi:10.3390/ijms26041519_

Round 1

Reviewer 1 Report

Comments and Suggestions for Authors

In this review, Martinotti and colleagues present the state of the art on the multiple properties and applications of propolis, which suggest for this natural resource a large-scale therapeutic potential given the anti-inflammatory and antioxidant capacities.

The text is organized systematically and in clear and precise language. The chapters are addressed in a thorough and structured way.

Minor points of improvement

I would add a chapter or a subchapter of the two main ones dedicated to the antioxidant role of propolis. Also in light of the fact that in figure 1, the topic is represented among the central ones of the review. Also in view of the use of this substance as an adjuvant in improving sports performance as happens for flavonoids of plant origin.

Author Response

In this review, Martinotti and colleagues present the state of the art on the multiple properties and applications of propolis, which suggest for this natural resource a large-scale therapeutic potential given the anti-inflammatory and antioxidant capacities.

The text is organized systematically and in clear and precise language. The chapters are addressed in a thorough and structured way.

We thank the reviewer for his/her appreciation of our manuscript.

Minor points of improvement

I would add a chapter or a subchapter of the two main ones dedicated to the antioxidant role of propolis. Also in light of the fact that in figure 1, the topic is represented among the central ones of the review. Also in view of the use of this substance as an adjuvant in improving sports performance as happens for flavonoids of plant origin.

We have inserted a new chapter (3. Anti-oxidant properties).

Reviewer 2 Report

Comments and Suggestions for Authors

In this review, the authors describe several pharmacological properties of propolis. The paper is generally well written, and the data presented are of important to the basic and applied fields of drug development as well as for nutraceutical application. The scientific writing is very good and easy to follow. 

A point of concern is why the authors do not include an illustration or a Table summarizes the major components that contribute to the pharmaceutical properties of propolis. 

Other points are:

-       The authors provided an excellent review of literatures about anti-inflammatory properties and related diseases of propolis. However, they mentioned that the medicinal properties of propolis are greatly influenced by the extraction method, i.e. solvent used. Thus, instead of pointing out that “propolis” show so and so, they should tell the reader what EXTRACT been used for the specific anti-inflammatory action reported in the study.

-       The reference 54 (Sake, J.M. ….) looks like something missing. Doi # is also missing.

Author Response

In this review, the authors describe several pharmacological properties of propolis. The paper is generally well written, and the data presented are of important to the basic and applied fields of drug development as well as for nutraceutical application. The scientific writing is very good and easy to follow. 

We thank the reviewer for his/her appreciation of our manuscript.

A point of concern is why the authors do not include an illustration or a Table summarizes the major components that contribute to the pharmaceutical properties of propolis. 

We thank the reviewer for this point.

We have now inserted a new Table (table 1) summarizing the major components of propolis.

Other points are:

-       The authors provided an excellent review of literatures about anti-inflammatory properties and related diseases of propolis. However, they mentioned that the medicinal properties of propolis are greatly influenced by the extraction method, i.e. solvent used. Thus, instead of pointing out that “propolis” show so and so, they should tell the reader what EXTRACT been used for the specific anti-inflammatory action reported in the study.

We thank the reviewer for this point. In fact, when the propolis extraction method was available in the literature we have now indicated it.

-       The reference 54 (Sake, J.M. ….) looks like something missing. Doi # is also missing.

We have amended this reference (now reference number 63).